# Bystander intervention is associated with reduced early mortality among injury victims in Cameroon

Kathleen O'Connor[1]*, Matthew Driban[1], Rasheedat Oke[2], Fanny Nadia Dissak-Delon[2], Signe Mary Magdalene Tanjong[3], Tchekep Mirene[4], Mbeya Dieudonne[3], Thompson Kinge[3], Richard L. Njock[5], Daniel N. Nkusu[6], Jean-Gustave Tsiagadigui[4], Cyrille Edouka[5], Catherine Wonja[5], Zachary Eisner[7], Peter Delaney[8], Catherine Julliard[2], Alain Chichom-Mefire[9], S. Ariane Christie[2]

1 University of Pittsburgh School of Medicine, Pittsburgh, Pennsylvania, United States of America, 2 Program for the Advancement of Surgical Equity (PASE), Department of Surgery, University of California Los Angeles, Los Angeles, California, United States of America, 3 Limbe Regional Hospital, Limbe, Cameroon, 4 Edea Regional Hospital, Edea, Cameroon, 5 Hopital Laquintinie de Douala, Douala, Cameroon, 6 Catholic Hospital of Pouma, Pouma, Cameroon, 7 University of Michigan School of Medicine, Ann Arbor, Michigan, United States of America, 8 Department of Orthopaedic Surgery, Cleveland Clinic, Cleveland, Ohio, United States of America, 9 Department of Surgery, Faculty of Health Sciences, University of Buea Faculty of Health Sciences, Buea, Cameroon

* koconnor990@gmail.com

**Data Availability Statement:** We have attached all data, de-identified of patient information, from the Cameroon Trauma Registry, used in this analysis as a Supporting Information file.

## Abstract

Despite high injury mortality rates, Cameroon currently lacks a formal prehospital care system. In other sub-Saharan African low and middle-income countries, Lay First Responder (LFR) programs have trained non-medical professionals with high work-related exposure to injury in principles of basic trauma care. To develop a context-appropriate LFR program in Cameroon, we used trauma registry data to understand current layperson bystander involvement in prehospital care and explore associations between current non-formally trained bystander-provided prehospital care and clinical outcomes. The Cameroon Trauma Registry (CTR) is a longitudinal, prospective, multisite trauma registry cohort capturing data on injured patients presenting to four hospitals in Cameroon. We assessed prevalence and patterns of prehospital scene care among all patients enrolled the CTR in 2020. Associations between scene care, clinical status at presentation, and outcomes were tested using univariate and multivariate logistic regression. Injury severity was measured using the abbreviated injury score. Data were analyzed using Stata17. Of 2212 injured patients, 455 (21%) received prehospital care (PC) and 1699 (77%) did not receive care (NPC). Over 90% (424) of prehospital care was provided by persons without formal medical training. PC patients were more severely injured (p<0.001), had markers of increased socioeconomic status (p = 0.01), and longer transport distances (p<0.001) compared to NPC patients. Despite increased severity of injury, patients who received PC were more likely to present with a palpable pulse (OR = 6.2, p = 0.02). Multivariate logistic regression adjusted for injury severity, socioeconomic status and travel distance found PC to be associated with reduced emergency department mortality (OR = 0.14, p<0.0001). Although prehospital injury care in Cameroon is rarely performed and is provided almost entirely by persons without formal

**Funding:** This work was supported by the National Institutes of Health (NIH 1K01TW012689-01 to SC, NIH R21TW010453 to CJ). The funders had no role in study design, data collection and analysis, decision to publish, or preparation of the manuscript.

**Competing interests:** The authors have declared that no competing interests exist.

medical training, prehospital intervention is associated with increased early survival after injury. Implementation of LFR training to strengthen the frequency and quality of prehospital care has considerable potential to improve trauma survival.

## Introduction

Low- and middle-income countries (LMICs) bear 90% of the world's mortality burden due to injury [1]. Prehospital care is a critical component of the survival chain following injury, but prehospital care is extremely limited in LMICs [2]. In May 2023, the 76th World Health Assembly (WHA) resolved that emergency, critical and operative care services are an integral part of a comprehensive national primary health care approach and foundational for health systems to effectively address emergencies [3]. Integrated prehospital and inpatient emergency care may address up to 54% of all-cause mortality across LMICs [2, 4, 5].

Basic prehospital life support may equal development of advanced interventions in terms of improving clinical outcomes. In settings without existing prehospital infrastructure, the World Health Organization (WHO) has recommended training bystanders as Lay First Responders (LFRs) as the first step toward prehospital infrastructure development [6]. LFR programs train non-medical professionals with high exposure to injury in basic principles of trauma prehospital care [4]. Trainee populations have included mototaxi drivers, taxi drivers, police officers, and firefighters [7–9]. LFR programs have been piloted in sub-Saharan Africa, but, to date, evaluations have largely focused on educational outcomes, program cost effectiveness, and reported utilization [8, 10–12]. The paucity of prospective clinical data collection in these settings has limited formal evaluation of the relationship between bystander intervention and clinical trauma outcomes.

Since 2015, the Cameroon Trauma Registry (CTR), a prospective, ongoing, multisite trauma registry, has collected data on over 30,000 injured Cameroonian patients, including data on prehospital care [12]. As the first step toward development of a data-driven LFR program for the Cameroonian context, the objective of this study was to 1) characterize existing clinical and prehospital care patterns in Cameroon and 2) to evaluate the association between bystander prehospital care and key clinical outcomes.

## Methods

### Study setting

Cameroon is a Central African nation with a population over twenty-seven million and no established prehospital care system. Ambulance access is prohibitively expensive and concentrated in its largest cities and use is generally limited to transport of established patients between hospitals rather than emergency response [6]. Ambulance drivers are not trained in medical care. The current practices for bringing injury victims to the hospital in the absence of formal prehospital infrastructure is unexplored in Cameroon.

### Cameroon Trauma Registry

We extracted data from a prospective multicenter cohort study of all trauma patients enrolled in the Cameroon Trauma Registry (CTR) in 2020. This included patients presenting to Laquintinie Hospital of Douala, Regional Hospital of Limbe, Catholic Hospital of Pouma, and Regional Hospital of Edea. These hospitals serve patients in both urban and rural settings and

represent public and private institutions. Inclusion criteria were patients of all ages presenting to the hospital for injuries who subsequently were admitted to the hospital, died in the emergency department, left against medical advice, or were transferred to another hospital. Patients were excluded if they were discharged directly from the emergency department. Patients enrolled in the registry following verbal consent were followed by trained research assistants from the emergency department to discharge. Data were collected separately from hospital records, initially on paper, and later were entered electronically into a secure REDCap database.

Data extracted from the CTR included demographics, prehospital care receipt and parameters, injury characteristics including injury severity, clinical patterns, treatments, and outcomes. Patients who received any kind of prehospital care (PC) were compared to patients who did not receive prehospital care (NPC). Prehospital care was defined as any attempted medical intervention or stabilization prior to presentation to the hospital, including care administered by either formally or informally trained providers. Care at another medical institution prior to arriving at a CTR was not considered to be prehospital care. Data regarding prehospital care was provided upon presentation by the patient, triage staff, transporter to hospital, and/or witness to injury. Prehospital care providers self-identified upon hospital presentation as bystanders, individuals involved in the accident, patient family/friends, or police/military. Prehospital interventions assessed were bleeding control (including compression and elevation), fracture immobilization, tourniquet placement, C-spine immobilization, topical burn treatment, backboard support, CPR, assistance to the recovery position, and provision of IV fluids or medicines. Highest estimated abbreviated injury severity was used to categorize injury severity.

## Ethics statement

Ongoing ethical approval for the CTR is maintained by The University of California Los Angeles and Cameroonian Ministry of Public Health. Prior to enrollment in the CTR, local research assistants secured informed consents from patients or surrogates using an IRB-approved script. (UCLA IRB#19–000086 and Cameroon Ministry of Public Health 2018/09/1094/CE/CNERSH/SP) Research assistants encouraged patients and surrogates to ask questions. Participation in the CTR is completely voluntary and participation did not effect medical care. As in many contexts with variable literacy, a standard verbal consent was administered to all patients with written documentation of consent status. Patients were excluded from CTR enrollment if they did not consent to participate. All patient information was de-identified at time of analysis.

## Reflexivity statement

Our longstanding collaboration is led by our Cameroon-based PI, Dr. Alain Chichom-Mefire. The authors would like to emphasize the core mission of building capacity of African junior scientists in our collaboration. This project established the knowledge base to support development of an evidence-based lay-first responder pilot that will be led by Cameroonian scientists Dr. Frida Embolo. It will also be used to support the prospective pilots and thesis work of postdoctoral fellow Dr. Elvis Tanue and the PhD of Dr. Vanessa Tabe.

## Data analysis

Data were summarized using means and standard deviation for normally distributed numeric variables and by medians and interquartile ranges for nonparametric numeric variables. Frequencies and proportions were reported for categorical variables. Comparisons between

groups were made using Chi squared (no tail) for categorical variables and Kruskal-Wallis analysis for numerical variables. We evaluated associations between prehospital care and patient outcomes using adjusted multiple logistic regression. For all analysis, alpha level of 0.05 was used for significance. Statistical analysis was conducted in Stata 17.

## Results

Overall, 2212 patients were enrolled in the CTR during the study period. The majority (1300, 76%) of injured patients were male, the median age was 31 (IQR 24,42), 88% (2033) lived in urban areas, and 95% (2087) had family access to a cellphone. Road traffic injuries were most common injury mechanism (1580, 74%) and patients most often presented to the hospital setting with closed fractures (479, 22%), hematomas (340, 16%), and deep lacerations (334, 15%). The median travel distance to the hospital following injury was 10 km (IQR 5,13). (Table 1)

### Prehospital care and transport patterns

Overall, 459 (20%) patients received prehospital care compared to 1699 (76%) who did not receive prehospital care. The vast majority (424, 93%) of prehospital care was provided by persons without formal training. Specifically, bystanders provided prehospital care for 70% (305), followed by relatives or friends (101, 23%) others involved in the accident (9, 2%), and those who self-identify as having medical training (9, 2%). The most common prehospital care interventions performed included bleeding control (370, 57%), fracture immobilization (139, 21%), and tourniquet placement (78, 12%). (Table 2)

### Characteristics of prehospital care recipients

There were no significant differences in sex or age between PC and NPC. PC patients were more frequently from rural areas (54% vs. 45% p<0.001) and did not have access to cellphones (91% vs. 95% p = 0.003) but were users of Liquid Petroleum gas (55% vs. 44%, p = 0.003), a proxy for increased socioeconomic status. PC patients reported greater travel distance to the hospital NPC 15km15km±28.3 vs. PC 17km17km±34.9 p<0.001). Prehospital care was significantly more common among patients transported by motorcycle taxi than by automobile taxis (31% vs. 5%, p<0.001). Just 20% of individuals transported by ambulance had prehospital care. (Table 1)

Additionally, PC more frequently presented with penetrating injury mechanisms (70% vs. 52% p<0.001), including gunshot or stab wounds and both superficial and deep lacerations. PC recipients had increased severity of injury by highest estimated abbreviated injury score (PC 3.2±0.88 vs. NPC 2.7±0.80) (p<0.001). (Table 1)

### Clinical consequences of prehospital care

Compared to NPC, the PC cohort presented with increased heart rate (PC 91bpm±14.5, vs. NPC 87bpm±18.5 p = 0.0001), reduced respiratory rate (PC 22±4.4 vs. NPC 23±10.3, p<0.001) and systolic blood pressure (PC 123mmHg±20.6 vs. NPC 127mmHg±23.8, p<0.001). Prehospital care recipients had increased rates of external bleeding (93% vs. 75%, p<0.001), increased rates of abnormal breath sounds (4% vs. 1% p = 0.001), and more abnormalities on primary survey overall (93% vs. 79%, p<0.001). However, the PC cohort had significantly increased rates of palpable pulse on presentation (99.8 vs. 98.6 p = 0.042). PC also had lower rates of severely depressed GCS (GCS<9 3% vs. 7% for NPC, p = 0.003). (Table 3) Multivariate logistic regression adjusted for injury severity identified PC cohort to be associated with increased injury survival. (OR 0.14, p<0.001)

**Table 1. Demographics of trauma patients and prehospital care.**

| | | No prehospital care | | Prehospital care | | P |
|---|---|---|---|---|---|---|
| | | Freq. | % | Freq | % | |
| **Sex** | Male | 1300 | 77 | 349 | 20.6 | 0.95 |
| **Age (IQR)** | | 32 (24, 42) | | 31 (25, 43) | | *<0.01* |
| **Urban/Rural** | Rural | 119 | 44.7 | 143 | 53.8 | *<0.01* |
| | Urban | 1570 | 81.4 | 310 | 16.1 | |
| **Cellphone** | No | 77 | 4.5 | 37 | 8.1 | *<0.01* |
| | Yes | 1616 | 95.1 | 417 | 91.6 | |
| **LPG*** | No | 864 | 50.6 | 201 | 49.2 | *<0.01* |
| | Yes | 835 | 44.2 | 254 | 55.8 | |
| **Travel (km) IQR** | | 10 (5, 12.3) | | 10 (5, 18) | | *<0.01* |
| **Injury mech** | RTI | 1269 | 78.2 | 311 | 19.2 | *<0.01* |
| | Stab/cut | 131 | 67.2 | 59 | 30.3 | |
| | Fall | 143 | 86.1 | 19 | 11.5 | |
| | Struck by person or object | 85 | 81 | 17 | 16.2 | |
| | Other | 25 | 73.5 | 9 | 26.3 | |
| | Gun | 10 | 38.5 | 15 | 57.7 | |
| | Scald | 3 | 27.3 | 8 | 73.7 | |
| | Burn | 5 | 55.6 | 4 | 44.4 | |
| **Injury type** | Closed frac | 399 | 83.3 | 80 | 16.7 | *<0.01* |
| | Hematoma | 329 | 96.8 | 11 | 3.2 | |
| | Deep lac | 240 | 71.9 | 94 | 28.1 | |
| | Open frac | 220 | 82.1 | 48 | 17.9 | |
| | Superficial lac | 125 | 61.9 | 77 | 38.1 | |
| | Stab | 74 | 77.9 | 21 | 22.1 | |
| | Bruise | 68 | 93.1 | 5 | 6.9 | |
| | Degloving | 50 | 72.5 | 19 | 27.5 | |
| | Sprain | 36 | 97.3 | 36 | 2.7 | |
| **Injury location** | Extremities | 555 | 83.8 | 107 | 16.2 | *<0.01* |
| | Face | 246 | 93.2 | 18 | 6.8 | |
| | Head/neck | 519 | 91.9 | 46 | 8.1 | |
| | Chest | 136 | 48.6 | 144 | 51.4 | |
| | Abdomen | 82 | 70.7 | 34 | 29.3 | |
| | Pelvis | 61 | 60.4 | 40 | 39.6 | |
| | Spine | 50 | 52.1 | 46 | 47.9 | |
| **Injury severity** | Missing score | 23 | 60.5 | 15 | 39.5 | *<0.01* |
| | Minor | 24 | 96 | 1 | 4 | |
| | Moderate | 534 | 93.7 | 25 | 4.4 | |
| | Serious | 957 | 77.2 | 250 | 20.2 | |
| | Severe | 124 | 43.4 | 152 | 53.2 | |
| | Critical | 21 | 65.6 | 10 | 31.3 | |
| | Unsurvivable | 16 | 88.9 | 2 | 11.1 | |
| **Transport type** | Taxi | 1061 | 94.7 | 60 | 5.3 | *<0.01* |
| | Private car | 208 | 53.5 | 181 | 46.5 | |
| | Mototaxi | 233 | 68.9 | 105 | 31.1 | |
| | Other | 54 | 44.7 | 67 | 55.3 | |
| | Police | 90 | 95.7 | 4 | 4.3 | |
| | Ambulance | 36 | 78.3 | 10 | 21.7 | |

*(Continued)*

**Table 1.** (Continued)

| | | No prehospital care | | Prehospital care | | P |
|---|---|---|---|---|---|---|
| | Walk | 14 | 33.3 | 28 | 66.7 | |
| | Unknown | 3 | 100 | 0 | 0 | |

* LPG = liquid petroleum gas, marker for increased socioeconomic status; all categorical variables reported as proportions and all numerical variables reported as medians with interquartile range

## Discussion

As the first step toward development of a data-driven LFR program for the Cameroonian context, the objective of this study was to 1) characterize existing clinical and prehospital care patterns in Cameroon and 2) to evaluate the association between bystander prehospital care and key clinical outcomes. We demonstrate that prehospital care is uncommon in Cameroon and is currently provided mostly by untrained scene bystanders. Although provision of prehospital care is significantly higher among patient with severe injuries, receipt of prehospital care is associated with decreased early mortality after injury, despite the fact that currently this care is largely provided by persons without formal training in first aid and safe transport. The efficacy of even informal prehospital intervention strongly suggests that implementation of sustainable prehospital care infrastructure, such as a formal LFR program, could be extremely impactful in improving trauma outcomes in Cameroon.

**Table 2. Classification of prehospital care providers and type of care provided.**

| | | Freq. | % |
|---|---|---|---|
| **Provider description** | | | |
| | Bystander | 305 | 70.4 |
| | Relative/friend | 101 | 23.3 |
| | Person involved | 9 | 2.07 |
| | Medic | 9 | 2.07 |
| | Other | 7 | 1.6 |
| | Police | 1 | 0.2 |
| | Unknown | 1 | 0.2 |
| | Driver | 0 | 0 |
| **Care provided** | | | |
| | Control bleeding | 370 | 56.7 |
| | Fracture immobilization | 139 | 21.3 |
| | Tourniquet placement | 78 | 11.9 |
| | C-spine immobilization | 37 | 5.6 |
| | Topical burn treatment | 12 | 1.8 |
| | Back board | 11 | 1.7 |
| | Other | 4 | 0.6 |
| | CPR | 1 | 0.15 |
| | Recovery position | 1 | 0.15 |
| | Unknown | 1 | 0.15 |
| | IV fluids | 0 | 0 |

The most common means of transporting patients to the hospital was commercial vehicles including 52% by taxi (1152) and 15% by motorcycle taxi (339). Only 3% (57) were brought to the hospital by ambulance. Of those brought to the hospital by ambulance, 78% did not receive prehospital care. (Table 1)

**Table 3. Clinical consequences of prehospital care.**

| | | No prehospital care | | Prehospital care | | P |
|---|---|---|---|---|---|---|
| | | Mean | | Mean | | |
| **Vitals** | SBP* | 128 | | 120 | | *<0.01* |
| | HR** | 87 | | 91 | | *<0.01* |
| | RR*** | 20 | | 22 | | *<0.01* |
| | Temp | 36.9 | | 36.8 | | *0.13* |
| | | **Freq** | **%** | **Freq** | **%** | |
| **Airway** | Not patent | 31 | 2 | 4 | 1 | *0.16* |
| | Yes patent | 1659 | 98 | 448 | 99 | |
| **Breathing** | No chest rise | 17 | 1 | 1 | 0 | *0.27* |
| | Abnormal chest rise | 1644 | 98 | 443 | 96 | |
| | Normal chest rise | 30 | 1 | 8 | 4 | |
| | Abnormal breath sounds | 28 | 98 | 19 | 96 | *<0.01* |
| | Normal breath sounds | 1664 | 2 | 450 | 4 | |
| **Circulation** | No palpable pulse | 23 | 1 | 1 | 0 | *0.04* |
| | Yes palpable pulse | 1669 | 99 | 449 | 100 | |
| | No external bleeding | 408 | 24 | 30 | 7 | *<0.01* |
| | Yes external bleeding | 1274 | 76 | 419 | 93 | |
| **GCS** | ≤9 | 120 | 7.06 | 15 | 3 | |
| | >9 | 1579 | 93 | 440 | 97 | |
| | | | | | | *<0.01* |
| **Overall** | No abnormalities | 363 | 21 | 34 | 7 | |
| | Yes abnormalities | 1336 | 79 | 421 | 93 | |

* systolic blood pressure

** heart rate

*** respiratory rate

Prior impact evaluation of prehospital care programs in Sub-Saharan Africa has been limited by lack of robust clinical data collection infrastructure. Our study builds on the existing literature by establishing the association between prehospital care receipt, key physiologic parameters (including heartrate and blood pressure), and emergency department survival. Although, these findings should be formally tested in other LMIC, we hypothesize that similar associations between bystander care and trauma survival will ultimately be demonstrated across contexts. These data also underscore the primacy of developing and maintaining prospective data collection capacity as a first essential step in LMIC surgical systems development, particularly in settings where the clinical environment does not maintain standard electronic records. In our collaborations experience, this first step is critical but extremely time and effort consuming, and publishers and funders can often be dismissive of early "descriptive" studies. However, the capacity to collect granular data is the only way to secure a data pipeline capable of identifying care gaps and supporting and evaluating systems development. Further development of ongoing research capacity, and support for this development, is needed across nations without reliable prehospital care infrastructure.

The injury and prehospital patterns identified in this study directly inform planning efforts for prehospital system development. To maximize access to prehospital care, LFR trainees ideally should have high exposure to injuries. As road traffic injuries constitute the injury mechanism for nearly three quarters of the patients in our cohort, and commercial vehicles transport over half of injured patients, our findings suggest taxi and mototaxi drivers potentially

represent an ideal target for LFR training in Cameroon. Mototaxi drivers have been trained as first responders elsewhere in sub-Saharan Africa, including Uganda, Chad, Nigeria, and Sierra Leone, with associated evidence of increase in access to prehospital care [8–11]. However, cultural context varies greatly between settings and buy-in from LFR trainees is critical to intervention feasibility. For these reasons, formal assessment of the acceptability of participating in LFR training among commercial drivers is a critical next step in program development, and is currently underway in Cameroon.

A formal review of trauma deaths by the National Cameroonian Trauma Quality Improvement Committee identified that hemorrhagic shock was key contributor to the nearly 80% of potentially preventable trauma deaths [13]. Here we present data demonstrating that the most common intervention performed by untrained prehospital responders was bleeding control, however, tourniquets were almost never applied. Rapid hemorrhage control with tourniquets for extremity hemorrhage in patients not yet in shock is strongly associated with reduced mortality [14]. Taken together, these data suggest that formal instruction on hemorrhage management has the potential to increase effectiveness of prehospital care providers and should be emphasized in a Cameroonian-tailored layperson first responder program [14].

Periodic review of clinical data from the CTR will remain critical to collaborating with local stakeholders for prehospital process planning. We plan on using the data to track rates of prehospital care, evaluate clinical outcomes, and analyze patterns of care among high-impact responders with future program implementation. As previously described, ongoing next steps driven by the data presented here include qualitative stakeholder interviews to best assess the acceptability of a commercial-driver lay-first responder trainee cohort and iterative adaptation of the LFR curriculum for targeted contextually-appropriate management of injury in Cameroon.

## Limitations

This study has several notable limitations. Due to the observational study design, we cannot infer causality in the relationship between prehospital care and trauma deaths. Furthermore, data on prehospital care receipt is contingent self-reporting by the patient or persons accompanying the patient. Consequently, prehospital care status may be more likely to be missing among the most critically injured patients, who may be less able to communicate these details.

## Conclusions

In Cameroon, prehospital care is uncommon and mostly provided by untrained bystanders but is associated with reduced early trauma mortality. Prehospital and clinical patterns can be used to tailor a LFR program for the Cameroonian setting. Commercial drivers provide most transport from the prehospital scene of injury to definitive care and represent a cohort with high situational exposure to injury. Given high rates of hemorrhage among injury victims, evidenced-based hemorrhage management should form a cornerstone of future LFR curricula. Finally, LFR program co-implementation in Cameroon with an existing trauma registry is critical to appropriately evaluate program impact and facilitate ongoing quality improvement to optimize care.

## Supporting information

**S1 Data. File containing deidentified data with all variables from the Cameroon Trauma Registry used in this manuscript's analysis.**
(XLSX)

## Acknowledgments

Alain Chichom-Mefire and S. Ariane Christie are joint senior authors. We appreciate our Cameroonian partners, research assistants and Zachary Eisner and Peter Delaney for their support and advisement.

## Author Contributions

**Conceptualization:** Kathleen O'Connor, Zachary Eisner, Peter Delaney, Catherine Julliard, Alain Chichom-Mefire, S. Ariane Christie.

**Data curation:** Kathleen O'Connor.

**Formal analysis:** Kathleen O'Connor, Matthew Driban, S. Ariane Christie.

**Investigation:** Kathleen O'Connor.

**Methodology:** Kathleen O'Connor.

**Project administration:** Rasheedat Oke, Fanny Nadia Dissak-Delon, Signe Mary Magdalene Tanjong, Tchekep Mirene, Mbeya Dieudonne, Thompson Kinge, Richard L. Njock, Daniel N. Nkusu, Jean-Gustave Tsiagadigui, Cyrille Edouka, Catherine Wonja, Alain Chichom-Mefire.

**Supervision:** Catherine Julliard, Alain Chichom-Mefire, S. Ariane Christie.

**Writing – original draft:** Kathleen O'Connor.

**Writing – review & editing:** Kathleen O'Connor, Fanny Nadia Dissak-Delon, Zachary Eisner, Peter Delaney, Alain Chichom-Mefire, S. Ariane Christie.

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
