## [Decision Letter · Decision Letter 0]

19 Feb 2024

PGPH-D-24-00069

Bystander intervention is associated with reduced mortality among injury victims in Cameroon

Dear Dr. OConnor 

Thank you for submitting your manuscript to PLOS Global Public Health. After careful consideration, we feel that it has merit but does not fully meet PLOS Global Public Health’s publication criteria as it currently stands. Therefore, we invite you to submit a revised version of the manuscript that addresses the points raised during the review process.

The paper addresses a crucial public health issue concerning trauma and prehospital care in low- and middle-income country (LMIC). However, reviewers have pointed out a lack of detail, particularly regarding the methods. It should be explicitly stated that the study included prehospital care provided by bystanders. Additionally, if patients transported via ambulances received any prehospital care and from whom, this information should be clarified. The quality and specifics of prehospital care provided by bystanders should also be detailed..==============================

We look forward to receiving your revised manuscript.

Kind regards,

Uzma Rahim Khan

Academic Editor

Journal Requirements:

1. Please amend your detailed online Financial Disclosure statement. This is published with the article. It must therefore be completed in full sentences and contain the exact wording you wish to be published.

a) State the initials, alongside each funding source, of each author to receive each grant. For example: "This work was supported by the National Institutes of Health (####### to AM; ###### to CJ) and the National Science Foundation (###### to AM)."

2. Please update your online Competing Interests statement. If you have no competing interests to declare, please state: “The authors have declared that no competing interests exist.”

3. Please upload your main article file as a .doc, .docx or .rtf file.

Additional Editor Comments (if provided):

Reviewers' comments:

Reviewer's Responses to Questions

**Comments to the Author**

1. Does this manuscript meet PLOS Global Public Health’s publication criteria? Is the manuscript technically sound, and do the data support the conclusions? The manuscript must describe methodologically and ethically rigorous research with conclusions that are appropriately drawn based on the data presented.

Reviewer #1: Yes

Reviewer #2: Yes

2. Has the statistical analysis been performed appropriately and rigorously?

Reviewer #1: Yes

Reviewer #2: Yes

3. Have the authors made all data underlying the findings in their manuscript fully available (please refer to the Data Availability Statement at the start of the manuscript PDF file)?

Reviewer #1: Yes

Reviewer #2: Yes

4. Is the manuscript presented in an intelligible fashion and written in standard English?

Reviewer #1: Yes

Reviewer #2: Yes

5. Review Comments to the Author

Reviewer #1: There is no data so far published regarding bystanders as this registry in Cameroon is going on since 2015 local context is missing .

STUDY SETTING : Tis must explain the settings /description of the locations/hospitals or organizations that are taking care of the registry or record . Study setting elaborates the broad description of setting that is giving you study population , hence remains unclear here.

DISCUSSION:

Please compare results with existing literature, acknowledge limitations, and suggest future research directions. Emphasize the study's significance, discuss unexpected results, and critically evaluate strengths and weaknesses. Conclude by summarizing key points and addressing ethical considerations. Adapt the discussion to the publication's requirements, ensuring a logical and well-organized flow.

Reviewer #2: EMS is a public health priority. Training lay providers in places without EMS is a practical method to develop EMS rapidly. Limited efforts have been made to leverage these networks, so this is an important study adding to the literature gap.

Methods:

Study Setting

Describe the study setting for the reader to understand the context. (City/village/state/province).

Cameroon Trauma Registry details:

Overview of the hospitals: How many hospitals? Type of hospitals (public, private)? (tertiary, secondary)? Patient population catered to (urban/rural)?

Briefly explain the trauma registry methodology to better understand the data collection. Who collected the data? Where did they collect the data (Emergency department, wards)? Was data extracted from medical records as well? Data collection timing/duration? Was the data collected digitally or paper based? Who calculated the Injury Severity Score?

Lines 157-159 are repeated.

Pre-hospital care: how was the medical information for prehospital care collected? Who gave history? Driver, transporters, bystanders may not witness the entire scene?

Results

The % for PC and NPC are different in abstract (line 72) and results section (line 199)

Pre-hospital care: What was the first referral facility? Were all these patients brought directly to the hospitals included in the study or were they first taken elsewhere and then brought later after stabilization?

Multivariate analysis

It would be interesting to see the multivariate results adjusted for injury type, mode of transportation, first responder.

Discussion:

The authors need to discuss their results and findings especially focusing on the bystanders.

6. PLOS authors have the option to publish the peer review history of their article (what does this mean?). If published, this will include your full peer review and any attached files.

**Do you want your identity to be public for this peer review?** For information about this choice, including consent withdrawal, please see our Privacy Policy.

Reviewer #1: **Yes: **Sumia Andleeb Abbasi

Reviewer #2: **Yes: **Natasha Shaukat

---

## [Decision Letter · Decision Letter 1]

27 Mar 2024

PGPH-D-24-00069R1

Bystander intervention is associated with reduced mortality among injury victims in Cameroon

Dear Dr. OConnor,

Thank you for submitting your manuscript to PLOS Global Public Health. After careful consideration, we feel that it has merit but does not fully meet PLOS Global Public Health’s publication criteria as it currently stands. Therefore, we invite you to submit a revised version of the manuscript that addresses the points raised during the review process.

We look forward to receiving your revised manuscript.

Kind regards,

Uzma Rahim Khan

Academic Editor

Journal Requirements:

Additional Editor Comments (if provided):

Reviewers' comments:

Reviewer's Responses to Questions

**Comments to the Author**

1. If the authors have adequately addressed your comments raised in a previous round of review and you feel that this manuscript is now acceptable for publication, you may indicate that here to bypass the “Comments to the Author” section, enter your conflict of interest statement in the “Confidential to Editor” section, and submit your "Accept" recommendation.

Reviewer #1: (No Response)

Reviewer #2: All comments have been addressed

2. Does this manuscript meet PLOS Global Public Health’s publication criteria? Is the manuscript technically sound, and do the data support the conclusions? The manuscript must describe methodologically and ethically rigorous research with conclusions that are appropriately drawn based on the data presented.

Reviewer #1: Yes

Reviewer #2: Yes

3. Has the statistical analysis been performed appropriately and rigorously?

Reviewer #1: Yes

Reviewer #2: Yes

4. Have the authors made all data underlying the findings in their manuscript fully available (please refer to the Data Availability Statement at the start of the manuscript PDF file)?

Reviewer #1: Yes

Reviewer #2: Yes

5. Is the manuscript presented in an intelligible fashion and written in standard English?

Reviewer #1: Yes

Reviewer #2: Yes

6. Review Comments to the Author

Reviewer #1: This topic and information is of great importance in field of trauma and injury.

However, the title the aim of study and conclusion are loosing consistency and coherence. Clear study objective statement needed to be mentioned for clarity.

Revisions have been made still discussions needs improvement in terms of reflection/reasoning and evidence relevance.

"Bystanders pre hospital intervention" This essence is lost from discussion till conclusion. It is very important and innovative to highlight bystanders intervention and investment in this area in terms of capacity building /raising awareness or life skills/first aid. If stidy largely aims to emphasise it's importance than essence shouldn't be lost. However, rest of things have been improvised well in revised version. Grammar and English can be revised before re submission.

Reviewer #2: (No Response)

7. PLOS authors have the option to publish the peer review history of their article (what does this mean?). If published, this will include your full peer review and any attached files.

**Do you want your identity to be public for this peer review?** For information about this choice, including consent withdrawal, please see our Privacy Policy.

Reviewer #1: **Yes: **Sumia Andleeb Abbasi

Reviewer #2: **Yes: **Natasha Shaukat

---

## [Decision Letter · Decision Letter 2]

4 Jun 2024

PGPH-D-24-00069R2

Bystander intervention is associated with reduced early mortality among injury victims in Cameroon

Dear Dr. OConnor,

Thank you for submitting your manuscript to PLOS Global Public Health. After careful consideration, we feel that it has merit but does not fully meet PLOS Global Public Health’s publication criteria as it currently stands. Therefore, we invite you to submit a revised version of the manuscript that addresses the points raised during the review process.

Thank you for this work based on the Cameroonian trauma registry that focuses on the potential of pre-hospital care (and by implication lay responder training) in improving trauma outcomes (emergency department mortality). Excellent revisions have been applied to the manuscript, and the authors have improved the text significantly from the minor revisions required. From a structure and content standpoint, the manuscript looks good and about set to go.

It is good practice to include the ethics approval numbers e.g (2018/09/1094/CE/CNERSH/SP for Le Comite National d’Ethique de la Recherche pour la Sante Humaine (CNERSH), and the University of California Los Angeles (IRB#19-000086) etc. *in the body of the manuscript *within the Ethics section for clarity.

Header rows of the tables should be written in full (even if they fit 2 lines each) so that they are stand-alone in a sense; as NPC, PC etc are not clear within the table (even if this has been described within the text). A superscript on the abbreviations on the table connected to a footer where these abbreviations are written in full is also an option.

As first, second, corresponding, and positionally last author (co-senior authorship acknowledged) are from the global north on this Cameroon focused project, it will be advisable for the authors to submit a reflexivity statement to show how they developed local junior researcher capacity by this study, and perhaps how they hope to include more junior researchers in leading writing subsequently, in the spirit of global health research equity promoted by Plos Global Public Health.

Beyond that, I believe that this important work should be published as soon as is possible. Thank you for your excellent revisions.

We look forward to receiving your revised manuscript.

Kind regards,

Barnabas Tobi Alayande

Academic Editor

Journal Requirements:

2. We have noticed that you have uploaded Supporting Information files, but you have not included a list of legends. Please add a full list of legends for your Supporting Information files after the references list.

Additional Editor Comments (if provided):

Reviewers' comments:

Reviewer's Responses to Questions

**Comments to the Author**

1. If the authors have adequately addressed your comments raised in a previous round of review and you feel that this manuscript is now acceptable for publication, you may indicate that here to bypass the “Comments to the Author” section, enter your conflict of interest statement in the “Confidential to Editor” section, and submit your "Accept" recommendation.

Reviewer #2: All comments have been addressed

2. Does this manuscript meet PLOS Global Public Health’s publication criteria? Is the manuscript technically sound, and do the data support the conclusions? The manuscript must describe methodologically and ethically rigorous research with conclusions that are appropriately drawn based on the data presented.

Reviewer #2: Yes

3. Has the statistical analysis been performed appropriately and rigorously?

Reviewer #2: Yes

4. Have the authors made all data underlying the findings in their manuscript fully available (please refer to the Data Availability Statement at the start of the manuscript PDF file)?

Reviewer #2: Yes

5. Is the manuscript presented in an intelligible fashion and written in standard English?

Reviewer #2: Yes

6. Review Comments to the Author

Reviewer #2: (No Response)

7. PLOS authors have the option to publish the peer review history of their article (what does this mean?). If published, this will include your full peer review and any attached files.

**Do you want your identity to be public for this peer review?** For information about this choice, including consent withdrawal, please see our Privacy Policy.

Reviewer #2: No

---

## [Editor Report · Decision Letter 3]

21 Jun 2024

Bystander intervention is associated with reduced early mortality among injury victims in Cameroon

PGPH-D-24-00069R3

Dear Ms. OConnor,

We are pleased to inform you that your manuscript 'Bystander intervention is associated with reduced early mortality among injury victims in Cameroon' has been provisionally accepted for publication in PLOS Global Public Health.

Best regards,

Barnabas Tobi Alayande

Academic Editor

All suggested changes have been applied as appropriate. Thank you.